# OpenReview forum: "EchoPrompt: Instructing the Model to Rephrase Queries for Improved In-context Learning"
_ICLR.cc/2024/Conference — ICLR 2024 Conference Withdrawn Submission_

### Official Review · Reviewer_KwXV · 2023-10-27

**Soundness:** 3 good
**Presentation:** 4 excellent
**Contribution:** 3 good
**Rating:** 6
**Confidence:** 4

**Summary:**

In this paper, the authors introduce a new prompting technique that decomposes conditional natural language generation (NLG) in two subtasks: first repeating the query and then generating the answer. Across a wide range of NLG tasks, testing a set of publicly available as well as closed language models, the authors show the superiority of the proposed approach over existing prompting techniques. An additional ablation study tries to investigate the main factor contributing to the performance gain, showing that the main benefits are due the query rewriting task acting as a query augmentation technique.

**Strengths:**

I find the main strength of the paper in its semplicity and in the extensive set of experiments conducted, showing a reasonable amount of evidence that EchoPrompt can effectively improve performance over widely adopted baseline on a set of commonly used benchmarks. The analysis, which I personally find still preliminary yet the most interesting feature of the paper, successfully disentangle some aspects of the performance gain. The connection to query augmentation is especially useful as it could help in devising new ideas to design additional prompting techniques. Additionally, as shown in the paper, EchoPrompt is complementary to the reasoning-based techniques like Chain of Thought.

**Weaknesses:**

I do not find any major weaknesses in the work. I think the main one, admittedly minor, is the preliminary analysis on the effectiveness of EchoPrompt. Additional analysis work, moving some results to the appendix, would have made the paper stronger. For instance, do the multiple rephrases bring lower performance even when doing query augmentation instead of EchoPrompt? How robust is the model to wrongly rephrased sentences? Given the results on the robustness to irrelevant text I would assume this has a low impact but it is interesting to see the results. Additionally, an error analysis on the failure cases might also help better understanding the causes that make EchoPrompt successful. Please note that these are not fundamental experiments to grant acceptance of this paper in the conference, they are mostly the reason that made me give a the score I gave instead of a clear accept.

I appreciated the limitations section where the authors mentioned the lack of explanation for why EchoPrompt results in performance gains. Regarding the additional computational cost, could retrieval techniques be used to fetch similar sentences rather than ask the model to generate them. Assuming the main benefits come from augmentation, this could bring the same gains and possibly reduce latency/costs.

Finally, I consider the paper to be a interesting contribution in the field LM prompting. The experiments are well executed and the analysis, even though preliminary, provides interesting insights. Another strength of the contribution is its being complimentary to other prompting method. Overall, I'm in favor of seeing the paper included in the conference.


----------

MINOR

To better read the results, especially figure 4 but also in all the tables, a random baseline should be added. This is only mentioned in section 4 for Code-davinci-002. Additionally, please specify what metrics you use to evaluate your method on all the datasets presented.


What version of ChatGPT backbone model was used to generate rephrased sentences (Sec 2.2)? Please add it in the manuscript.


-------


TYPOS

Figure 2: repeatition -> repetition

In Zero-shot-CoT why was Zero capitalized?

Table 2, bottom left section extra quotation mark?

**Questions:**

Do the authors think RLHF might have an effect on performance when EchoPrompt is used? In the section on GPT-3.5-Turbo the sentence "After manual qualitative analysis, we observe that the model generates descriptive rather than instruction-based extractable answers" made me think that the effect of the different fine-tuning method on EchoPrompt ecould be further explored.

In Footnote 2: I did not grasp why quotation marks won't work for understanding when query rephrasing was completed? Couldn't they be used in the same way they are used for repetition?

The description of results in table 3, 4, 5, 6 are presented without always presenting the setup. I reconstructed that table 3 is few-shot setup while table 4,5, 6 are 0-shot. I'd suggest to make the tables self-contained and adding this info, as well as making clear at the begining of each paragraph presenting the results.

Although the robustness to irrelevant text results are understandable, it'd be useul to quickly explain the features of the GSMIC-4k dataset.

Even though the commonsense reasoning task is mentioned, its results, and Table 8 showing them, are never referenced in the manuscript.

---

### Official Review · Reviewer_avEG · 2023-10-31

**Soundness:** 3 good
**Presentation:** 4 excellent
**Contribution:** 2 fair
**Rating:** 3
**Confidence:** 4

**Summary:**

This paper proposes to change the generic Chain of Thought (CoT) prompt string "Let's think step by step" to phrases like "Let's repeat/reiterate/restate the question and also think step" (EchoPrompt). The method can also be used
in few-shot settings and in settings with no CoT, where it simply asks for a question rephrasing. The goal is to let the model reformulate the initial question, and the hypothesis is that this will lead to better results than standard prompting and standard generic CoT. The experimental comparisons show some modest gains and some modest losses relative to standard generic CoT. A set of ablation studies suggests that the performance of EchoPrompt is driven by the rephrased question.

**Strengths:**

The paper reports on a lot of experiments, and the ablation studies are thoughtful.

**Weaknesses:**

1. The proposal itself has very little theoretical substance. It seems good in general to look for precedence in the cognitive or developmental literature, but I am not sure of the hypothesis that would link this work to that of Joseph and Joseph and Ross (cited on page 1). The protocols in that work are different in so many ways that the comparison seems misleading to me.

2. Since the proposal is so simple, the weight of the paper rests on the results. In my view, the gains from this method seem very modest and are likely to be overshadowed by other innovations in in-context learning and/or fine-tuning.

3. Comparing Table 1 with Figure 4/Table 9 suggests to me that minor rephrasings of the EchoPrompt string can lead to larger performance changes that EchoPrompt achieves over its baselines. This leaves me worried that the method is exploiting unknown and largely unknowable features of Instruct fine-tuning rather than capturing something deep about how in-context learning works.

**Questions:**

1. Table 9's caption says that the method "It outperforms the prior state of the art in numeric reasoning". What datasets is this referring to precisely, and what is the presumed state of the art?

2. Is Table 9 reporting test-set results or dev-set results? This matters since dev and test performance for some of these tasks is often far apart.

---

### Official Review · Reviewer_KWdy · 2023-10-31

**Soundness:** 3 good
**Presentation:** 3 good
**Contribution:** 2 fair
**Rating:** 5
**Confidence:** 4

**Summary:**

The paper proposed a new prompt design strategy, namely EchoPrompt, which aims to paraphrase the input query before answering. Based on this idea, the authors have designed various paraphrasing strategies, including compound sentences, putting question first, short and simple sentences, and repetition. All strategies need no human efforts and rely the model themselves to generate the new phrase. For evaluation, the authors consider both few-shot and zero-shot settings on various tasks, such as numerical reasoning, logical reasoning, reading comprehension, and commonsense reasoning. The results demonstrate great improvements.

**Strengths:**

1. The paper is easy to follow.

2. The idea is simple yet effective.

3. Experiments demonstrate the performance gain.

**Weaknesses:**

1. The main contribution of the paper lies in the rephrase idea in prompt. Although the idea is simple yet effective, the authors fail to provide in-depth insights as the techniqical contribution. Most of the experiments just rephrase the performance values in the table.

2. The main reason leading to the improvements is still unclear. Does the rephrase enhance the understanding process or it is just various forms of questions, which increase the probabilities of the model answering correctly? What about a baseline that first generates many rephrase queries, then answers each query separately, and finally obtains the results via an ensemble method?

**Questions:**

see weaknesses

---

### Official Review · Reviewer_oXYq · 2023-11-03

**Soundness:** 2 fair
**Presentation:** 3 good
**Contribution:** 2 fair
**Rating:** 3
**Confidence:** 3

**Summary:**

This paper introduces a new way of prompting LLMs in which the LLMs are explicitly prompted to generate a version of the query it has been asked before solving it. This means that e.g. in zero-shot chain-of-thought (COT) prompting, the prompt changes from ”Let’s think step by step” -> “Let’s repeat the question and also think step by step”. This idea can be extended to the few-shot setting as well. However, the in-context demonstration needs to be augmented with a “reformulation” of the question as well. This is done with automatically with the help of ChatGPT. The authors experiment with a few ways to generate the rephrase of the query (e.g. rephrase to compound sentences, rephrase to short and simpler sentence, rephrase to putting the question first, and just repeat verbatim). The authors test their idea on multiple LLMs and on several mathematical and logical reasoning benchmarks

**Strengths:**

**Originality**

Unfortunately the contributions of the paper are not original. The paper proposes a very minor incremental idea without much intuition about why the proposed method would work. Therefore the paper is not strong wrt originality

**Quality**

The paper runs experiments with multiple LLMs on multiple benchmarks and observes decent improvements in most.
In the space of LLM prompting, there are potentially unbounded number of options to try, however the paper does a decent job trying a few options of query rephrasing

**Clarity**

The paper is clearly written and fairly easy to follow and should be accessible to anyone reading it.

**Significance**

Given the missing intuition behind the proposed method, the paper is unfortunately not very significant.

**Weaknesses:**

* In an introductory paragraph where the idea is motivated, the paper states that current prompting techniques can suffer from logical errors, symbol mapping issues, and omission of intermediate steps. It is unclear whether the proposed solution by the authors solves any of the above problems that  they have stated to motivate the work.
  * Why do you think asking the LLM to repeat the question would lead to the reduction of logical errors, symbolic mapping issues and omission of intermediate steps? Its cool that it works in the benchmarks that it has been tested on, but what is the inductive bias of the model that would make it work on other unseen tasks.
* Similar to above point - The paper would significantly benefit if it sheds some light on why it works? I acknowledge that the authors have stated in the limitation section that they are not sure why the model works but unfortunately that might not be good enough for ICLR. I encourage the authors to investigate the reasoning behind the better performance. Is it more computation because of the repeated question? Or is it something else?
* Figure 3: The LLM outputs - “Re-writing in simple words”. Can you tell what about the re-writing is simple in this case?

**Questions:**

* The paper will significantly improve if the authors investigate and find out why the method works and then use that to motivate the proposed approach.